# Surely but quickly: Quick convergence using Self Sampling Attention

## Abstract

Transformer is a very versatile architecture used in fields like computer vision, natural language processing, and audio signal processing. However, its data-hungry nature and the quadratic complexity of the Self-Attention(SA) layer make it slow and costly to train. Solutions like self-attention alternatives and distillation aim to reduce the computational complexity of training Transformers, but convergence remains costly and challenging. As shown by Liu et al. (2023), the amplification effect makes it hard for a Transformer using traditional SA mechanism to find the attention quickly. Therefore, in this work, we propose GauTransformer, a dual-phase learning model. In the first phase, it uses a stochastic approach based on sampling the key tokens with learned Gaussian distributions to find optimal attention maps faster than the standard Self-Attention layer. In the second phase, it accelerates transformer convergence through distillation. We demonstrate that the GauAttention module can be a powerful mechanism to achieve competitive performance while decreasing the computational cost of training. Furthermore, in many settings, we empirically show that GauAttention can reduce the training time to half the number of steps compared to the traditional method of training transformer architectures.

## 1 Introduction

Since its inception, Transformers have revolutionized the field of Deep Learning (DL), notably in Computer Vision (CV) (Dosovitskiy et al., 2020), Natural Language Processing (NLP) (Vaswani et al., 2017) and audio tasks (Koutini et al., 2021). The model quickly proved itself to be an architecture of choice for many, potentially replacing Recurrent Neural Networks (RNNs) for NLP tasks. In CV, Dosovitskiy et al. (2020) proposed ViT, a Transformer-based image classifier that outperformed Convolutional Neural Networks (CNNs) on the ImageNet benchmark (Deng et al., 2009). These compelling results can be partly credited to the Self-Attention mechanism.

Despite the model's numerous prowess, its adoption still remains inaccessible for many because of several drawbacks. First, since the Self-Attention (SA) layer compares each pair of features (tokens) within a training example, its complexity becomes quadratic in relation to the number of tokens. This can lead to significant slowdowns when processing long sequences of words in NLP, such as language modeling with Wikitext-300 (Merity et al., 2016) dataset, or when dividing images into numerous patches for classification, as seen with ImageNet (Dosovitskiy et al., 2020; Liu et al., 2021). Additionally, machine learning practitioners and researchers have found that Transformers are slow to converge (Bachlechner et al., 2021; Liu et al., 2023) and data-hungry (Wang et al., 2022), making them costly to train and potentially impractical for use cases where data is scarce. Furthermore, these training deficiencies contribute to increased use of GPUs and other resources, leading to negative environmental impacts due to high energy consumption and $CO_2$ emissions (Chien et al., 2023). For instance, training models such as ChatGPT, the popular Transformer-based chatbot, leave the same carbon footprint of around 262 round trip flight from Munich to New York (Truhn et al., 2024).

Researchers have been proposing solutions to address the high computational complexity in training, for instance, Reformer (Kitaev et al., 2020), Flash Attention (Dao et al., 2022) and Linformer (Wang et al.,

2020). The idea of these methods mostly relies on the introduction of SA alternatives to tackle the problem. Other work like DeiT (Touvron et al., 2021) incorporates distillation to a Vision Transformer to avoid the need for a larger pretraining dataset. Yet the amount of steps needed for a model to converge directly affects computational costs, thus discouraging the use of Transformers for smaller budgets or environmentally conscious applications. Despite being more than two dozens alternative approaches tackling its complexity (Tay et al., 2022), efforts aimed towards speeding up convergence of these models are limited to only a handful (Bachlechner et al., 2021; Kenneweg et al., 2023; Dai et al., 2022; Liu et al., 2023). In addition, these methods focus on new optimizers and pretraining rather than directly on the model's architecture. Therefore, we found it relevant to explore a technique that concentrates more on the architectural design of Transformer.

The motivation behind this work arises from the observation that the self-attention module assigns scores to all keys for each query, striving to find ideal attention scores. Instead of comparing every feature pair, we hypothesize that sampling a single key per query and optimizing the sampling distribution to emphasize relevant keys can accelerate the search for optimal attention.

To model this, we introduce the Gaussian Attention module (GauAttention), which samples tokens through a learnable Gaussian distribution to quickly identify important features for each training example. The GauTransformer comprises two phases: the first phase uses GauAttention to discover useful attention maps, while the second phase distills these maps into a standard Transformer. This Self-Sampling Attention mechanism enhances the quality of attention score selection, stabilizing training and reducing the number of required steps. By addressing the amplification effect that destabilizes early training stages (Liu et al., 2023), we aim to streamline the learning of optimal attention maps through the more efficient GauTransformer.

**Our main contributions can be summarized as follows:**

**(1)** Propose a dual-phase learning Transformer (GauTransformer) accelerating the standard Transformer's convergence rate, speeding up training and thus reducing computational costs.

**(2)** Introduce Gaussian-Attention (GauAttention): A stochastic alternative to Self-Attention that leverages Gaussian distribution-based sampling to efficiently learn the optimal key for each query. This approach mitigates the amplification effect, allowing it to converge faster than traditional Self-Attention during training.

**(3)** Show experimentally the effectiveness of the approach. Notably, the GauTransformer was able to speed up training more than twice on CIFAR-10 and three times on ImageNet.

## 2 Related Works

### 2.1 Efficient Transformers

Transformer is a succession of building block components: the encoder block, mainly characterized by the SA layer. Despite its demonstrated effectiveness in the aforementioned tasks, the Transformer architecture remains data hungry (Wang et al., 2022), impacted by an unstable training (Bachlechner et al., 2021; Liu et al., 2023), and burdened by the quadratic complexity of the SA layer, causing its training to be costly and slow (Dao et al., 2022). Consequently, researchers have formulated methods to optimize the model, focusing on three main approaches: **reducing attention complexity**, **knowledge transfer**, and **accelerating convergence**. This Section introduces topics related to ways of making Transformers more efficient.

**Reducing Attention complexity.** Reducing the complexity of SA significantly improves inference time for models with a large number of tokens. Research offers various methods to bypass full attention computation, each with its own speed/accuracy trade-offs. Examples include Performer (Choromanski et al., 2020), which reduces attention complexity to $\mathcal{O}(n)$ using an approximating kernel, Reformer (Kitaev et al., 2020), which reduces attention complexity to $\mathcal{O}(n \log n)$ using local attention, and FlashAttention (Dao et al., 2022), which reduces complexity by optimizing I/O operations, among many others.

**Knowledge Transfer.** Optimizing Transformer training can be achieved by using distillation from pre-trained networks, allowing the model to avoid learning attention maps independently. Touvron et al. (2021) proposed

using a distillation query token to transfer knowledge, achieving competitive results with ViT on ImageNet-1k without large-scale pretraining. Furthermore, TinyViT employs encoder block-wise feature distillation from larger Transformers to smaller ones, speeding up knowledge transfer and reducing inference time (Wu et al., 2022). To efficiently transfer its representations, the GauTransformer employs the same approach but with encoders different in nature and similar size(GauTransformer encoder to standard Transformer, see Figure 2).

**Accelerating Convergence.** Another way to make the training of Transformers more efficient is to speed up its convergence rate. This work focuses on improving convergence. The following section provides an overview of the current state of Transformer convergence rates.

## 2.2 Slow convergence of the Transformer

The convergence speed of a DL model is the amount of gradient steps needed for it to reach a local minimum. Xiao et al. (2021) have shown that the Transformer model's convergence is slower than its CNN counterpart by comparing RegNets (Xu et al., 2021) and ViTs (Dosovitskiy et al., 2020) with similar parameters counts. One probable cause of this identified by Bachlechner et al. (2021) is the vanishing gradient phenomena observed in Transformers due to their typically deep architectures (often dozens of encoder blocks, including many perceptions layers totaling millions and even billions of parameters) (Dosovitskiy et al., 2020; Chen et al., 2023). Despite vanishing gradients being a known cause of slow convergence (Han et al., 2021), according to Liu et al. (2023) in the specific case of Transformers, the source of the issue is rather produced by the Amplification effect. The Amplification effect is a phenomenon where the training is destabilized by the significant fluctuations of the Transformer encoder outputs between each gradients updates. This is caused by the encoder's reliance on its residual layer.

**Techniques to accelerate Convergence.** To speed up convergence in Transformers, researchers have proposed several innovations. One notable method is Adaptive Model Initialization (Admi) (Liu et al., 2023), which aims to stabilize the initial training steps through better weight initialization. Furthermore, Kenneweg et al. (2023) narrowed their efforts towards optimization by incorporating Armijo line search into the Adam optimizer (Kingma & Ba, 2017) and demonstrating that their approach decreases the amount of updates needed to converge compared to the standard optimizer. In the field of object detection, researchers have introduced the unsupervised pretraining method UP-DeTr (Dai et al., 2022), which halves the convergence epochs of DeTr (Carion et al., 2020), and the Spatially-Modulated Co-Attention technique (Gao et al., 2021), which reduces convergence steps to one-fifth of the original requirement.

Bachlechner et al. (2021) proposed a simple yet effective technique to accelerate convergence by adding zero-initialized parameters to the skip connection while introduced $\sigma$Reparam to address stability by preventing entropy collapse.

While effective, none of these approaches addresses slow convergence by improving the self-attention search process to identify optimal keys faster. Our Self-Sampling Attention mechanism aims to stabilize training by enhancing the quality of attention score selection, thereby reducing the number of training steps required. Given that the amplification effect aggravates instability in the early stages of training (Liu et al., 2023), our intuition is that distilling pretrained attention maps can mitigate this issue. Thus, we propose offloading the task of learning optimal attention maps to a more efficient module: the GauTransformer.

## 3 Gaussian Attention

This section introduces the motivation, intuition and architecture details behind the Gaussian Attention layer, a novel learning mechanism designed to enhance the efficacy of the search of optimal attention maps compared to a vanilla Transformer. Unique challenges posed by GauAttention are then addressed, along with a discussion of the convergence speedups achieved over traditional self-attention.

### 3.1 Preliminary definitions.

Let us consider a set of data containing $M$ samples $\mathcal{D} = \{(x_i, y_i)\}$ where $x_i \in \mathbb{R}^{W \times H \times C}$ are images with spatial resolution $W \times H$ and $C$ channels and $y_i \in \mathbb{N}$ is one of the $N$ class labels. In the training process of a Transformer-based image classifier, we aim to learn a parameterized function $t_\theta : \mathbb{R}^{W \times H \times C} \to \hat{y}_i$, where $\hat{y}_i$ is the predicted $c$ class label of the input image and $\theta$ the parameters vector. The considered loss function is the cross-entropy $(\mathcal{L}_{cls})$ between the predicted output of the classifier $\hat{y}_i$ and the class labels $y_i$, as in the following:

$$\mathcal{L}_{cls}(y, \hat{y}) = -\sum_{c=1}^{N} y_i^c \log(\hat{y}_i^c). \tag{1}$$

**Self-Attention.** As mentioned earlier, the Transformer's strength lies in its Self-Attention mechanism, which enables it to capture long-range relationships of its inputs. In this operation, every pair of features is compared and assigned scores of "importance" or "value". These scores, learned by the model's parameters, serve as a filter to emphasize important features and suppress noisy information (Vaswani et al., 2017). The Self-Attention layer comprises three linear layers: Query, Key, and Value. Values represent input features, queries identify features of interest, and keys are compared with queries to compute scores. As detailed in Eq. equation 2, the dot products between queries and keys yield scores, and the scores are normalized using softmax (Eq. equation 3) and dimension square-root division. Subsequently, values are multiplied by their corresponding scores and summed across queries. In Multi-headed attention, this process is repeated with distinct SA modules, followed by sequence-wise feature merging.

Given an input $x$, the SA process goes as follows : firstly, the layer applies attention to a sequence of tokens, therefore, in the case of ViT, $x$ is reshaped into a set of $p^2$ patches. From there, the module generates, through perceptrons, query $(Q)$, key $(K)$, and value $(V)$ matrices of dimensions of $p^2$ tokens of $d$ length (choose arbitrarily, often same as the patches of $x$ i.e. $\frac{C \times W \times H}{p^2}$ , in the case of ViT). The original self-attention formula is defined by the following:

$$\text{Attention}(Q, K, V) = \text{softmax}(\frac{QK^\top}{\sqrt{d}})V. \tag{2}$$

The softmax function used in the SA layer is defined by the following :

$$\text{softmax}(z) = \frac{e^{z_p}}{\sum_{j=1}^{K} e^{z_j}}. \tag{3}$$

### 3.2 Gaussian-Attention

For each query, the self-attention module learns to assign scores to all keys for every training example, aiming to find the ideal attention scores. As opposed to comparing each pair of features, we hypothesize that by sampling a single key per query and adjusting the sampling distribution to focus on relevant keys, we could accelerate the process of finding the optimal key.

We assume that important keys are distributed according to a Gaussian, and to model this, we propose the Gaussian Attention module (denoted as GauAttention).

The GauAttention layer process begins by assigning learnable parameters, denoted as $\mu_s \in \mathbb{R}^{\frac{W}{p} \times \frac{H}{p}}$ and $\sigma_s^2 \in \mathbb{R}^{\frac{W}{p} \times \frac{H}{p}}$, to each token of each input (images in our case), initialized to 0 and 1 respectively. During inference, we sample the Cartesian coordinates of each value using a Gaussian sampling (Grami, 2019). As the Gaussian sampling yields non-differentiable nodes, we use the reparametrization trick (Kingma & Welling, 2013) by standardizing the samples (denoted by $S$) with the learnable parameters $\mu_l$ and $\sigma_l$ as:

$$\mathcal{N}(z \mid \mu, \sigma^2) = \frac{1}{\sqrt{2\pi\sigma^2}} \exp\left(-\frac{(z-\mu)^2}{2\sigma^2}\right). \tag{4}$$

$$S \sim \frac{\mathcal{N}(\mu_s, \sigma_s^2) + \mu_l}{\sigma_l^2}. \tag{5}$$

Subsequently, we select the sampled coordinates of the values through differentiable grid sampling interpolation (Jaderberg et al., 2016). In short, the technique assists the network to select an index, that would normally be a discrete operation, in a differentiable manner. In our case, we approximate the sampled position of the values by applying a bilinear interpolation kernel as:

$$\mathcal{B}(z, s) = (s - \lfloor s \rfloor)z_{\lfloor s \rfloor} + (\lceil s \rceil - s)z_{\lceil s \rceil}, \tag{6}$$

where $z$ is a grid of vectors to which we want to select from and $\mathcal{B}$ is the differentiable bilinear interpolation. $s$ is the sampled position, $\lfloor s \rfloor$ and $\lceil s \rceil$ are respectively the floor and ceiling rounding operations defined as:

$$\lfloor s \rfloor = \text{round}(s - 0.5), \quad \lceil s \rceil = \text{round}(s + 0.5). \tag{7}$$

Finally, GauAttention can be formulated as:

$$\text{GauAttention}(V) = \mathcal{B}(V, S), \tag{8}$$

where $V$ are the values and $S$ are the standardized samples, as mentioned before. Similarly to Self-Attention, we apply the GauAttention multiple times in parallel across the token dimensions and concatenate the scores with different Gaussian parameters to allow the layer to explore multiple possible keys. The GauAttention layer is integrated into an encoder Transformer block and then used to learn the optimal attention maps. Visualization of the entire process is shown in Figure 1. Altogether, this process is done for each image and each layer, therefore each encoder block has its own set of Gaussian parameters for every image. Typically, learning parameters are exclusively part of the model architecture, using parameters directly for the data can potentially result in overparameterization and in unmanageable memory sizes in the cases of high volumes of data. In the following subsections **Phase** $A$ and **Compressions Networks** we detailed how those issues are addressed.

### 3.3  Dual-Phase Training

Since GauAttention aims to find optimal attention maps for each encoder block on a specific training set, it cannot be evaluated on unseen data, hence the necessity for a second phase of training. In spite of that, a condition is required to conclude the first training phase. Training a Transformer using GauAttention occurs in two phases: first, training the GauTransformer, followed by distilling the knowledge of the GauTransformer to the standard Transformer.

#### 3.3.1  Phase $A$: GauTransformer Training Process

Normally, a Transformer encoder block consists of an Attention Layer followed by a Feedforward network linked via a skip connection. The GauTransformer follows a similar architecture (see (a) in Figure 2). Performance on the training set is evaluated at each epoch, and if it falls within a pre-selected range, training stops. Considering the potential overparametrization on the data, caution must be taken not to set the upper bound too high to prevent overfitting, which could hinder representations for later distillation. As stated in the training details, we opt to stop the training at around 92% accuracy. This threshold was selected arbitrarily but was found empirically to be effective. Each training data point is encountered once per epoch, so not every parameter needs to be updated at each iteration. Consequently, a separate optimizer for Gaussian parameters updates only at epoch ends. Moreover, since gradient updates occur once per epoch, they can be executed on CPU to conserve GPU memory. In summary, the GauTransformer training process

should require fewer epochs to converge compared to its Self-Attention-based counterpart. This is because the sampling approach to identify key features avoids the noise associated with testing every feature using the scaled-dot-product, thus mitigating the amplification effect problem identified by Liu et al. (2023). A visualization comparing the selection of key process of Self-Attention vs. GauAttention can be seen in the **Understanding the Key Selection Process** section from the Experiments.

### 3.3.2 Compression Network: Tackling Challenges of Large Datasets

As the amount of data increases, the RAM memory requirements become an issue. For instance, in the case of ImagetNet-1k containing 1.3 million images with a resolution of 224 by 224, one Gaussian Attention layer with a patch size of 16 by 16 yields 196 tokens. For each image and each head, we have parameters for Cartesian coordinates (as we are sampling in 2D) for mean and standard deviation. The product of these dimensions totals approximately 25 billion parameters per encoder. In order to make this methodology feasible for bigger datasets, we have developed a compression network (a simple 4 layer MLP) responsible for generating the $\mu$ and $\sigma$ from smaller dimension embedding tokens for each image (see (b) in Figure 1). Experimental results demonstrate that the method compresses the Gaussian parameters effectively while adding minimal computation overhead. Using this approach, we were able to narrow down the Gaussian parameter count to 350 M parameters per encoder.

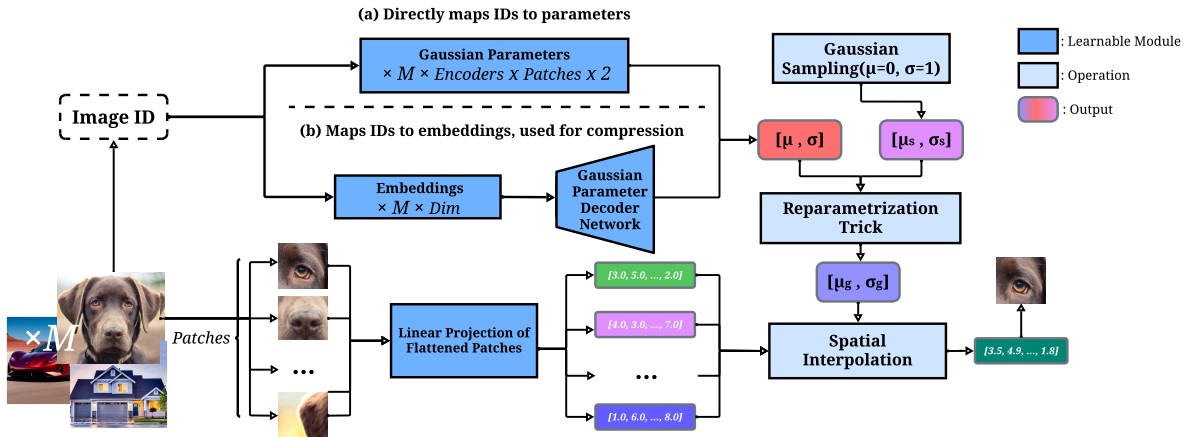

Figure 1: Diagram of phase $A$ of the GauTransformer training, where squared-corner shapes are training modules and rounded-corner ones are outputs. Dark blue modules have learnable parameters. (a) Here, each image ID is directly mapped to its saved set of Gaussian learnable parameters (b) In this case, each image ID is mapped to a smaller embedding that is then decoded by the Parameter Decoder Network to generate the Gaussian parameters. As stated previously, adopting the approach makes training a GauTransformer feasible on large datasets.

### 3.3.3 Phase $B$: Distillation Process

Once the GauTransformer is trained, it is time to incorporate its knowledge into a standard Transformer. Similarly to TinyViT (Wu et al., 2022), to distillate knowledge from one Transformer to another, we perform an encoder-wise feature distillation (Heo et al., 2019) as well as a soft distillation of the output logits.

The distillation is performed offline, allowing the attention maps from the teacher network (GauTransformer) to be saved, thus sparing computation. During validation, the model is evaluated the same way a vanilla Transformer would be, by simply measuring the student's performance on validation.

In the following equations, references to predictions of a full Transformer, using either Self-Attention or Gaussian-Attention, are denoted as $\mathcal{T}$ and $\mathcal{G}$ expressions, respectively, whereas the Transformer encoder function outputting the attention maps are identified using the $a$ suffix.

In Eq. equation 9, we describe the feature distillation loss as the KL-divergence between the attention maps from the Transformer and GauTransformer encoders:

$$\mathcal{L}_{feat}(x) = \sum_{i=1}^{F} (\mathcal{T}_a(x)_{i=1}^{F} \log(\frac{\text{softmax}(\mathcal{T}_a^i(x))/t}{\text{softmax}(\mathcal{G}_a^i(x))/t})), \tag{9}$$

where $F$ is the attention map feature dimension and $t$ is a temperature factor. Following, we have the Soft Distillation loss, which is simply the cross entropy between the soft labels (teacher predictions, in our case, the GauTransformer) and the predictions of the student (the vanilla Transformer):

$$\mathcal{L}_{soft}(x) = \mathcal{L}_{cls}(f(x), g(x)). \tag{10}$$

Finally, the full Gaussian Transformer loss $L_{Gau}$ is the weighted sum of each of the above-described loss functions. Each loss is multiplied by a weight factor to adjust their contribution at each gradient step. We set a fixed weight for the Feature Distillation Loss (Eq. equation 9), the Soft Distillation loss (Hinton et al., 2015) (Eq. equation 10) and Hard label loss from the object (Eq. equation 1), sum all the losses and update the Transformer at each step, described in the following Eq. equation 11:

$$\mathcal{L}_{Gau}(x, y, \hat{y}) = \alpha \mathcal{L}_{soft}(x) + \beta \mathcal{L}_{cls}(y, \hat{y}) + \gamma \mathcal{L}_{feat}(x), \tag{11}$$

where $\alpha$, $\beta$, and $\gamma$ are, respectively, the weights of the soft, hard, and feature losses.

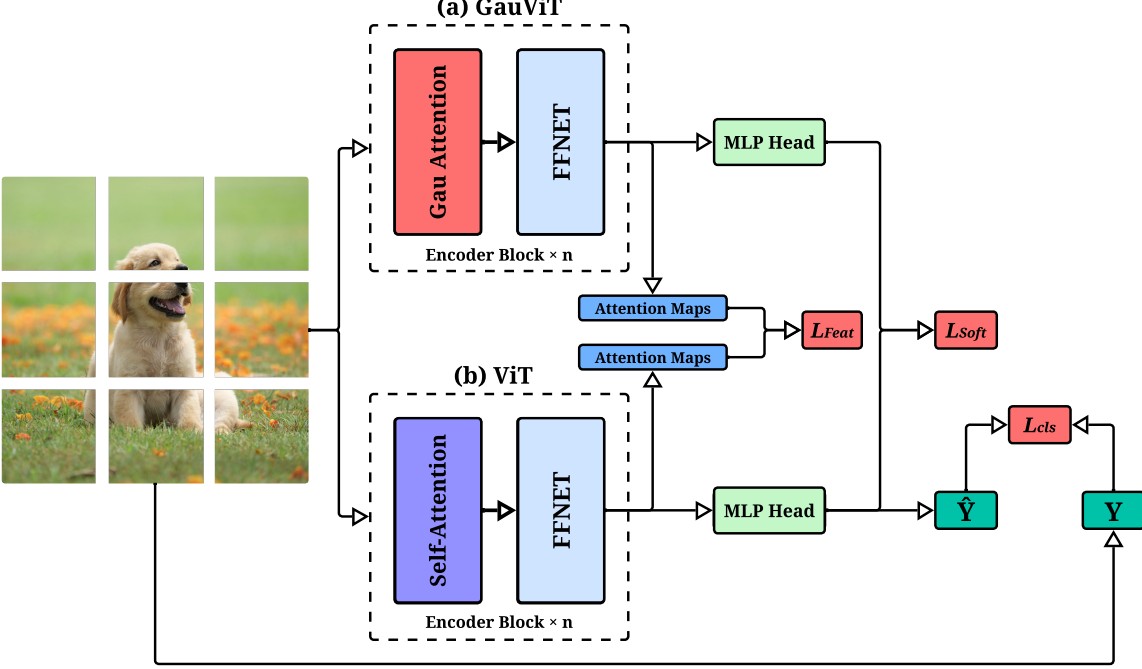

Figure 2: Diagram of the phase $B$ of the GauTransformer training: the distillation process of (a) GauTransformer to (b) Vanilla Transformer.

## 4 Experiments and Evaluation

In this section, we evaluate and compare the convergence speed of vanilla ViT (Dosovitskiy et al., 2020), ViT with ReZero (Bachlechner et al., 2021) (referred to as $ViT_{R0}$), ViT with $\sigma$Reparam (Zhai et al., 2023) (referred to as $ViT_\sigma$) and GauViT for image classification tasks.

### 4.1 Experimental settings

**Datasets.** For the experiments designed in this work, we selected to conduct the study on two traditional benchmarks for images: CIFAR-10 and ImageNet. The CIFAR-10 dataset, which contains $60,000$ images across 10 classes. For the ILSVRC-2012 ImageNet dataset, comprising 1.3 million images with $1,000$ classes. CIFAR-10 served as a proof of concept for the proposed method, and ImageNet was responsible for determining if Self Sampling Attention would perform well on more challenging benchmark. The image resolutions used for CIFAR-10 and ImageNet are 32x32 and 224x224, respectively.

**Metrics.** We evaluated the convergence speed by calculating the number of epochs needed to reach a performance plateau. Also, we evaluate the accuracy to know if the models are comparable with others, even though the main point is to have a similar performance with faster training. To ensure clarity, we define an epoch as one training cycle in which each example in the dataset has been used once for gradient updates of the neural network. In addition, we use cross-entropy loss to train our models and qualitatively visualize the speed of convergence. We will refer to different $GauViT$ trained in phases $A$ and $B$ as $GauViT_A$ and $GauViT_B$, respectively.

**Model variants.** For each datasets, two model sizes were used to make our experiments: Atto and Tiny for CIFAR-10 and ImageNet respectively. These names come from ConvNeXt size naming convention (Liu et al., 2022; Woo et al., 2023) by giving suffixes with matching models with similar parameter counts. The different model configurations can be seen in Table 1.

Table 1: Configurations of model scales used for experiments.

| Model Scale | Layers | Hidden Size | FF Size | Heads | Model Params |
|---|---|---|---|---|---|
| Atto | 7 | 384 | 384 | 12 | 6.3 M |
| Tiny | 8 | 768 | 768 | 12 | 29.9 M |

### 4.2 Experiments and Analysis

**Dual Training Setup.** As detailed in Section 3.3, the training of GauViT is conducted in two phases. For all ViT training, we use the same hyperparameters for fair comparison with all methods. All training is done on an NVIDIA A100 GPU using an AdamW optimizer (Loshchilov & Hutter, 2019) with a learning rate of $1 \times 10^{-4}$, betas (0.9, 0.999), and a weight decay of 0.03, along with a cosine linear warmup scheduler over 15 epochs. Additionally, we apply the following data augmentation techniques: RandAugment (Cubuk et al., 2019), RandomCrop, and RandomHorizontalFlip. We trained all models with a limit budget of 400 and 500 epochs for ImageNet and CIFAR-10 respectively. To speed up training for ImageNet, we used mixed precision. Typically, ViT training for ImageNet includes data augmentation methods such as CutMix (Yun et al., 2019) and Mixup (Zhang et al., 2018) for better performance. However, since attention maps are saved for each image, the use of these augmentations would have required a specific implementation. As the intent of this work is to measure the efficacy of our method, these powerful augmentations were omitted to ensure the performance improvements are indeed attributed to GauAttention.

As shown in Table 2, we perform a warmup-up (phase $A$) for GauViT, which, with only around 10 epochs, can reach over 90 of accuracy in the training set of both datasets with the different sizes of models. For both datasets, we stop training once the model reaches between 92% and 98% accuracy. This phase is responsible for learning the distribution of our learnable attention layer. As mentioned earlier, in the case of ImageNet we used an MLP with 4 layer of 512 dimensions to compress the learnable Gaussian parameters. For CIFAR-10 and ImageNet, the $\alpha$, $\beta$, and $\gamma$ used of the loss were respectively $[0.2; 0.1; 0.7]$ and $[0.85; 0.05; 0.1]$.

Table 2: Training results of GauViT's Phase $A$ on the training set for both datasets: CIFAR-10 and ImageNet.

| Model | Dataset | Epoch | Accuracy |
|---|---|---|---|
| GauViT$_A$-Atto | CIFAR-10 | 12 | 98.32 |
| GauViT$_A$-Tiny | ImageNet | 11 | 92.14 |

**Main Experiment.** We compare the convergence of all trained models in Table 3. For both datasets, the table is separated by the equivalent last 2 performances peak in terms of accuracy. Additionally, we show the graph of the cross entropy loss on the validation for all models on both datasets at Figure 3. The results reported for GauViT in table Table 3 include the epochs of the phase A.

GauViT demonstrates superior performance across all benchmarks, narrowly surpassing ViT$_{R0}$ (4.34 speedup) on ImageNet while showing notable efficiency gains by accelerating ViT by 4.56 times. ViT$_{R0}$ consistently ranks as the second-best model in all cases, highlighting its stability as a high-performing alternative. Interestingly, ViT$_\sigma$ shows minimal impact on CIFAR-10. These results position GauViT as both a performant and efficient choice, with ViT$_{R0}$ as a competitive fallback.

Table 3: Convergence Speed of different methods: ViT, ViT$_{R0}$, ViT$_\sigma$ & GauViT. The Atto configuration is used for CIFAR-10 and Tiny for ImageNet

| Dataset | CIFAR-10 | | | ImageNet | | |
|---|---|---|---|---|---|---|
| Model | Epoch | Accuracy | SpeedUp | Epoch | Accuracy | SpeedUp |
| ViT | 361 ± 30 | 83.39 | | 193 | 64.04 | |
| ViT$_{R0}$ | 282 ± 49 | 83.02 | ×1.28 | 83 | 64.12 | ×2.32 |
| ViT$_\sigma$ | 319 ± 36 | 83.23 | ×1.13 | 159 | 64.14 | ×1.21 |
| GauViT | **150 ± 9** | 83.08 | **×2.41** | 80 | 64.22 | **×2.41** |
| ViT | 450 ± 7 | 84.12 | | 374 | 65.05 | |
| ViT$_{R0}$ | 372 ± 61 | 84.12 | ×1.21 | 86 | 65.08 | ×4.34 |
| ViT$_\sigma$ | 444 ± 48 | 84.14 | ×1.01 | 163 | 65.20 | ×2.29 |
| GauViT | **165 ± 5** | 84.47 | **×2.72** | 82 | 65.28 | **×4.56** |

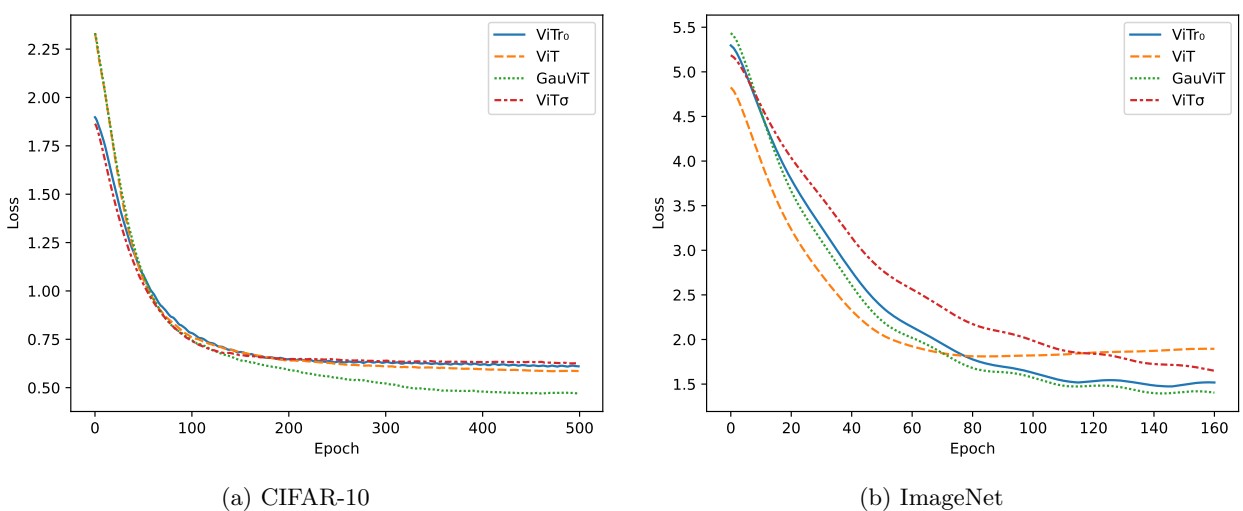

(a) CIFAR-10                    (b) ImageNet

Figure 3: Cross Entropy losses per epoch graphs on the validation sets of CIFAR-10 (a) and ImageNet (b) during training.

**Training time benchmark.** The training time of each method is compared in Table **??**. The benchmark is conducted on two datasets: CIFAR-10 and ImageNet. The reported training times include both the forward and backward passes for each batch in the training set. For GauViT, as it is trained in two phases, the total

training time for each phase is reported individually, with the sum presented in the bottom row. In the second phase, the output features from the teacher model used for distillation are saved to reduce additional computation. In the CIFAR-10 benchmark, GauViT achieved a significant reduction in training time, saving 3.2 hours to the standard training compared to ReZero, which saved only 1 hour. However, in the ImageNet benchmark, where the total number of epochs for GauViT and ReZero are similar (82 vs. 86, respectively), the computational overhead of GauViT's distillation phase resulted in a slightly slower total training time, exceeding ReZero's by 1.9 hours.

Table 4: Training time benchmarks on ImageNet and CIFAR-10. The ViT configuration Atto is used for CIFAR-10 and Tiny for ImageNet

| Dataset | ImageNet | | | CIFAR-10 | | |
|---|---|---|---|---|---|---|
| Model | Epoch Time (s) | No Epochs | Total Time (h)↓ | Epoch Time (s) | No Epochs | Total Time (h)↓ |
| ViT | $919 \pm 1$ | 374 | 95.5 | $44 \pm 3$ | 450 | 5.5 |
| ViT$_{R0}$ | $995 \pm 2$ | 86 | **23.8** | $44 \pm 2$ | 372 | 4.5 |
| ViT$_\sigma$ | $881 \pm 7$ | 163 | 39.9 | $55 \pm 1$ | 444 | 6.8 |
| GauVit$_A$ | $1061 \pm 9$ | 11 | 3.2 | $41 \pm 2$ | 12 | 0.1 |
| GauVit$_B$ | $1141 \pm 14$ | 71 | 22.5 | $46 \pm 3$ | 153 | 2.0 |
| GauViT | | 82 | 25.7 | | 165 | **2.1** |

**Ablation on loss.** As previously mentioned (Sec. 3.3.3), the optimization of GauViT is done using 3 different loss components: $\mathcal{L}_{soft}$ (Eq. equation 10), $\mathcal{L}_{cls}$ (Eq. equation 1) and $\mathcal{L}_{feat}$ (Eq. equation 9). To develop an intuition of the degree of importance the loss from the attention maps should have, an ablation on the sensitivity of each weight ($\alpha$, $\beta$ and $\gamma$) was conducted. All training runs were executed with GauViT-Atto on CIFAR-10 with 12 heads. The ablation in Table 5 reveals that in the range of 0.5 to 0.8 weights, the $\gamma$ weight does not seem to be the most determinant hyperparameter to optimize but for a fixed $\gamma$, in the majority of cases, having a $\beta$ loss weight higher than the $\alpha$ yields faster convergence.

Table 5: Ablation results on loss sensibility on CIFAR-10. All the reported training configurations are benchmarked until reaching a common peak accuracy of 84%. The emphasis on feature loss weight ranges from 50% to 80%. Subsequently, every possible combination of hard and soft label losses is tested in steps of 10%.

| Loss Weight | | | Convergence rate |
|---|---|---|---|
| $\gamma$ ($\mathcal{L}_{feat}$) | $\alpha$ ($\mathcal{L}_{soft}$) | $\beta$ ($\mathcal{L}_{cls}$) | Epoch |
| 0.5 | 0.1 | 0.4 | 168 |
| 0.5 | 0.2 | 0.3 | 171 |
| 0.5 | 0.3 | 0.2 | 165 |
| 0.5 | 0.4 | 0.1 | 192 |
| 0.6 | 0.1 | 0.3 | 164 |
| 0.6 | 0.2 | 0.2 | 166 |
| 0.6 | 0.3 | 0.1 | 172 |
| 0.7 | 0.1 | 0.2 | 151 |
| 0.7 | 0.2 | 0.1 | 168 |
| 0.8 | 0.1 | 0.1 | 161 |

**Understanding the Key Selection Process.** In order to better understand the key selection process of GauAttention as opposed to Self-Attention, the average selected query across all attention heads of a handful of images during training was measured. See Figure 4.

Qualitatively, we observe that with GauAttention, the important key is quickly identified, whereas with SA, the model displays a noisier selection process as training progresses. For instance, in the case of the image (I), both models ended up going for the key id $\approx 43$ but Vit went for other keys in the earliest epochs (key id 51 and 52). This may be a consequence of the amplification effect as identified by (Liu et al., 2023). The simpler learning of the scores in GauViT's case makes the training more stable.

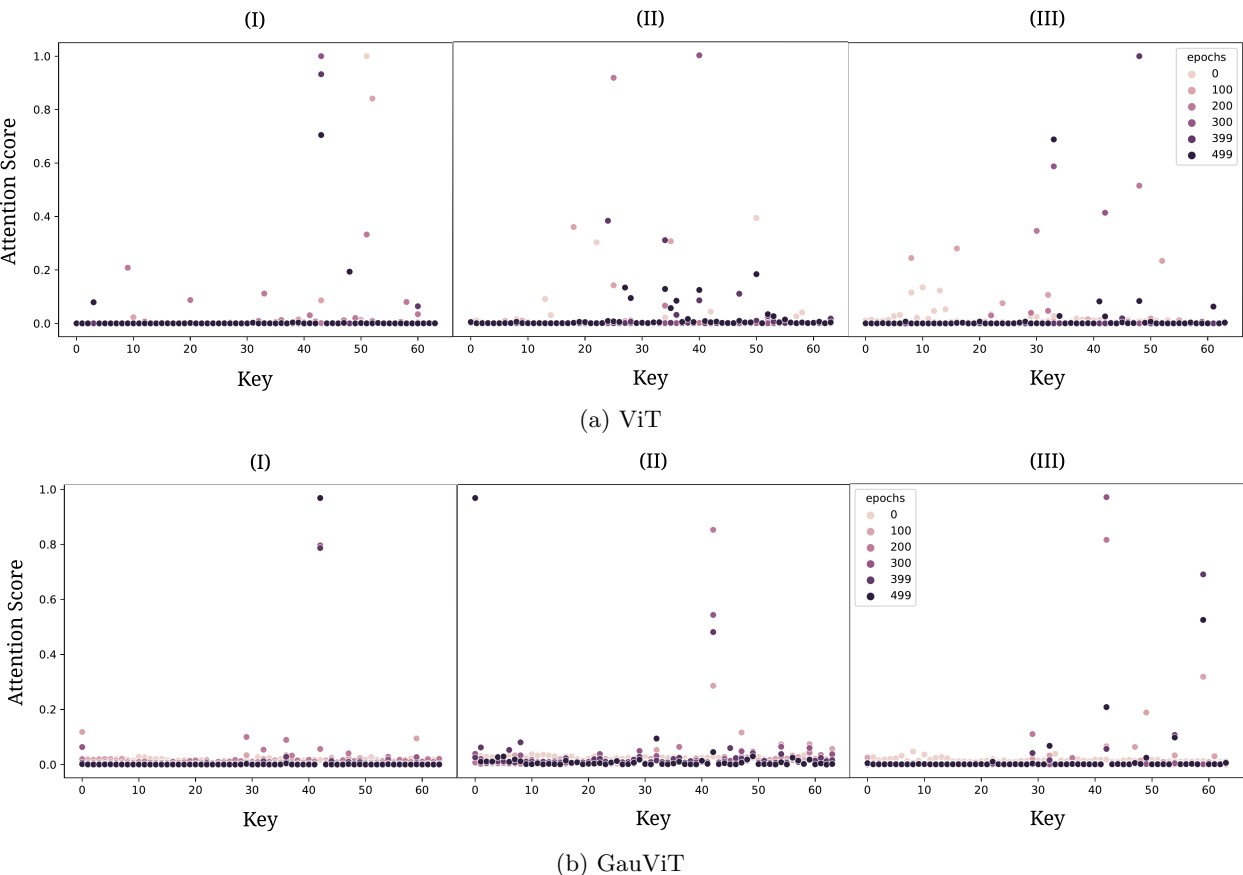

Figure 4: Attention scores attributed for each key, averaged across queries for a handful of images(identified as I, II and III). The scores are taken from training on CIFAR-10 from the first encoder block a (a) ViT and (b) GauViT.

## 5 Conclusion

This work introduced GauTransformer, a model aimed at improving the convergence rate and efficiency of standard Transformers. Our results demonstrate GauViT's superior performance across benchmarks, achieving a notable 4.56-fold acceleration over standard ViT on ImageNet and a 2.72-fold speedup on CIFAR-10. GauViT narrowly outperforms $ViT_{R0}$ on ImageNet while greatly improving performance on CIFAR-10, indicating that the Gaussian Attention layer effectively enhances the convergence rate by smoothing the attention search process. However, the implementation complexity, narrow performance margin on ImageNet, and slight computational overhead due to distillation make ReZero a more attractive option. In future work, we plan to find a way to reuse the GauAttention module to create a single-phase training pipeline, as well as exploring the application of the module on different applications and different Transformer architecture.

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
