# OpenReview forum: "Surely but quickly: Quick convergence using Self Sampling Attention"
_TMLR — Rejected by TMLR_

### Review · Reviewer_9tkK · 2024-09-21

**Summary Of Contributions:**

This work proposes the GauTransformer, an sturctural modification on the vanilla Transformer with a two-phase training strategy, to accelerate the convergence speed of Transformers. The modular contribution includes a distribution-based sampling embedding that helps the self-attention mechanism to learn the optimal keys of the tokens, as well as a compression network that generates low-dimensional embeddings to tackle the high memory requirements in large datasets. The two-stage strategy involves training GauTransformers to identify optimal attention maps quickly and distilling the knowledge from GauTransformers into a standard Transformer, which allows the model to be evaluated on unseen data. Empirical results demonstrate the proposed approach can increase 2~4x training speed while maintaining comparable performance with counterpart methods.

**Audience:**

Yes

**Broader Impact Concerns:**

None.

**Claims And Evidence:**

Yes

**Requested Changes:**

In addition to the evaluations and analyses mentioned above, there are also some suggestions regarding the current submission:

* To provide a more detailed explanation of the "mitigating the amplification effect problem," “the sampling approach to identify key features avoids the noise associated with testing every feature using the scaled dot-product "is not convincing.

* Could the author provide more implementation details of the compression network? It help to confirm the architectural contritions more clearly.
* It is encouraged that the author can provide the error bar of experimental results to exclude the influence of random training factors.

**Strengths And Weaknesses:**

**Strength**:

* The motivations are well-summarized: The paper highlights the practical challenges faced by Transformers, especially its data-hungry natures, slow convergence, and high computing costs, effectively underscoring the significance of the addressed problem.

* The paper provides good discussions with related works: Different from previous approaches taken to improve the Transformer's efficiency, including reducing attention's complexity, knowledge transfer, and accelerating convergence by optimizers. The proposed method focuses on both improving certain components and adopting the matched training strategies rather than solely addressing the one.


**Weakness**:


* About GauAttention: While the module can be innovative in some ways, it also leads to potential challenges, especially the overparameterization problem  (learnable mean and variance are introduced for each token). To compensate for the parameters, the paper proposes a series of additional designs, including training procedures and compressed networks. Although these designs can solve the problem, they may increase the complexity and training difficulty of the model in practice. This trade-off requires careful evaluation in more applications to determine whether this improved approach is worthwhile.
* Concern about the generalization performance: As the authors also notice the potential risks in the generalization performance degradation, I encourage the authors to evaluate the proposed method on more datasets (current two) and more Transformer models.

* The criterion of training stop: To overcome potential overfitting, the authors use a fixed accuracy threshold (e.g., 92%) during training as a condition for stopping training, which may seem too empirical and ad hoc. In different datasets or tasks, especially in non-classification tasks, this fixed threshold may not apply.

* About the performance: The two-stage training strategy does have limitations in practicability and application scenarios. According to the results of Table 3, although the two-stage training method can improve the training efficiency in some cases, the improvement in model performance may be insignificant. It suggests that the two-stage training method may need to be further optimized, such as an implementation of an end-to-end version or achieving a more significant improvement in results.
* About the weight configurations in Equation 11: From the observation of Table 4, the influence of weights seems not significant in convergence rate. And the relationship between convergence rate and weight configurations is not so clear. In addition, I encourage the author to set partial of the weight as zeros to identify whether each loss term has played an important role.

---

> ### Author Response · Authors · 2024-11-01
> **Response to Reviewer 9tkK**
>
> We thank the reviewer for finding that our motivation is well summarized in the paper and that our paper provides good discussions with related works.
>
> Let us address each requested change individually:
>
> Evaluation in more applications, dataset & Transformers
> Although we recognize the relevance of exploring more applications and Transformers, for the scope of this application, we planned to focus on Image Classification.
>
> Motivation behind Gau attention and Amplification effect
> We appreciate your feedback and acknowledge that we could have more clearly articulated the motivation behind our use of Gaussian sampling to optimize self-attention. In response, we have revised the motivation for GauAttention in the introduction, the conclusion of the related works, and in Section 3.2 of the Methodology.
> To address the 'amplification effect problem' in more detail: traditional self-attention mechanisms assess scores across all keys for each query, which can lead to substantial noise due to comparisons of less relevant features. Our approach, which samples a single key per query, reduces this noise by attempting to focus solely on the most relevant keys. This process allows for an accelerated and efficient determination of attention scores.
> Compression Network
> Thank you for pointing that out. To compress the Gaussian parameters, we simply used an MLP. Following the suggestion, we added this detail in the subsection “3.3.2 Compression Network: Tackling Challenges of Large Datasets” as well as the MLP configurations in the section “4.2 Experiments and Analysis”.
>
> Error bar
> In line with this suggestion, we trained three seeds for each model on CIFAR-10 to provide a more robust evaluation, as shown in Table 3. However, limitations in our training budget prevented us from doing the same for ImageNet.

---

### Review · Reviewer_KvKK · 2024-09-23

**Summary Of Contributions:**

The paper aims to improve the convergence speed of self attention transformers. They claim to mitigate the amplification effect studied by Liu et al. where the post-layer norm layer has high dependency on residuals. They propose a two phase pipeline to replace self-attention. In the first phase, the method learns one gaussian distribution per image for each attention head in the encoder. This would take too much memory, so a compression network that learns to predict $\mu, \sigma$ parameters from image embeddings is used. In the second phase, these learnt gaussian maps are distilled into a standard self-attention transformer by minimising KL divergence between the features, and cross-entropy between the soft-labels. Results on ViT image-classification training show 2.21-2.57x fewer iterations are required than the vanilla training on CIFAR-10, and 1.74-3.28x fewer iterations on ImageNet (this is worse than ReZero, the only baseline presented).

**Audience:**

No

**Broader Impact Concerns:**

None.

**Claims And Evidence:**

No

**Requested Changes:**

1. Critical - Please improve the motivation and description of the method to make it easier to understand and judge. For example, please add more details about the method in the introduction, especially the fact that it requires learning separate gaussian parameters for each sample.

2. Critical - Please report the time-taken per iteration for both of them, and also ReZero. I would also directly measure the time-taken instead of iterations when reporting the speedup.

3. Critical - Please add more baselines. These could be other works focused on stabilising convergence, such as [1-5] mentioned in the previous comment, or other types of self-attention alternatives already mentioned in related work. Also, the method description begins with the sampling performed during inference, and it's unclear without training details why this is important.

4. Critical - Please clearly and correctly describe the amplification effect, and why your method mitigates it (with empirical evidence, such as showing lower dependence of post LN layers on residuals).

5. Improvement - Please improve the related work section and verify the claims made. It would help to contextualise and compare your work to the mentioned papers, rather than just describing them. Please contextualise your work with respect to papers [1-5] cited in the previous comment.

6. Improvement - Results outside image classification would be helpful, such as language, to see if GauAttention really has the expressivity of self-attention.

**Strengths And Weaknesses:**

**Strengths**
1. An alternative to self-attention is proposed which is shown to require fewer iterations of training.

**Weaknesses**
1. The method is poorly described and motivated, with pieces of information scattered around with no clear rationales for why they were chosen. Overall, despite reading the method description multiple times, I am still not confident I understand it.

2. The convergence speed is measured using number of iterations. However, it is not obvious to me how one iteration of GauTransformer is comparable to vanilla training.

3. Lack of baselines/empirical validation of improvements. Only ReZero is benchmarked, and it outperforms the proposed method on ImageNet (the bigger out of two datasets reported).

4. The description of the amplification effect in the introduction is incorrect based on the cited paper. The Liu et al. paper specifically mentions unstable gradient updates is not the root cause of hard transformer training as that's fixed by adaptive optimisers. Instead, they say the root cause/amplification effect is the post LN layer having high dependency on residuals. Moreover, citation to the paper is wrong, as it was published in EMNLP 2020 but the paper’s citation says 2023. Since this is a central motivation of this work, it is important to discuss it properly. It is also unclear how their method mitigates or targets the amplification effect.

5. The related work section has a lot of issues. a) It says prior work mostly focuses on optimisers/pretraining rather than architectures, which is false. b) It also says prior work hasn't focused on improving convergence by modifying attention, which ignores papers like [1-5] cited below.

[1] Nguyen, Tam Minh, et al. "Improving transformers with probabilistic attention keys." *International Conference on Machine Learning*. PMLR, 2022.

[2] Nguyen, Tan, et al. "Improving transformer with an admixture of attention heads." *Advances in neural information processing systems* 35 (2022): 27937-27952.

[3] Nguyen, Toan Q., and Julian Salazar. "Transformers without tears: Improving the normalization of self-attention." *arXiv preprint arXiv:1910.05895* (2019).

[4] Zhai, Shuangfei, et al. "Stabilizing transformer training by preventing attention entropy collapse." *International Conference on Machine Learning*. PMLR, 2023.

[5] Gao, Peng, et al. "Fast convergence of detr with spatially modulated co-attention." *Proceedings of the IEEE/CVF international conference on computer vision*. 2021.

---

> ### Author Response · Authors · 2024-11-01
> **Response to Reviewer KvKK**
>
> We thank the reviewer KvKK for all the points mentioned, which were carefully taken into consideration to improve our work.
> Let us address each requested change individually:
>
> Enhance the method's motivation and clarity
> We appreciate your feedback and acknowledge that we did not clearly articulate the motivation for our use of Gaussian sampling to optimize Self-Attention.  We thank you for highlighting this oversight.
>
> We revised the motivation behind GauAttention in the introduction, the end of the related works, and Section 3.2 of the Methodology. To summarize, the GauTransformer is motivated by the observation that traditional self-attention assigns scores to all keys for each query. We propose that instead of comparing every feature pair, sampling a single key per query and optimizing the sampling distribution to focus on relevant keys can accelerate the search for optimal attention scores. We use parameters specific to each training example to rapidly determine the attention for that particular instance.
>
> Report the time taken per iteration
> We thank you for the suggestion, following the review, we added a training time benchmark table in our manuscript (Table 4).
> Include additional baselines, such as other works on convergence stabilization or self-attention alternatives, and clarify the significance of the sampling during inference without training details.
> We thank you for providing the papers 1 to 5 as additional references for our work, after carefully looking at the papers, we would like to remind the reviewer that our aim was to reduce the steps for convergence, not necessarily the complexity as it is the case for some of these papers referred. Let us address each paper individually:
> [1] Nguyen, Tam Minh, et al. "Improving transformers with probabilistic attention keys.", 2022. &
> [2] Nguyen, Tan, et al. "Improving transformer with an admixture of attention heads." (2022).
> These papers are very interesting. We note that [1] also incorporates Gaussian sampling into attention; however, both papers ([1] and [2]) focus on reducing the computational complexity of Transformers rather than improving their convergence rate.
> [3] Nguyen, Toan Q., and Julian Salazar. "Transformers without tears: Improving the normalization of self-attention." (2019).
> Although this paper is interesting, we struggled to find its relevance in our work as it studies normalizations at different stages of the Transformer’s encoder and compares their performance in different training resources regiment.
> [4] Zhai, Shuangfei, et al. "Stabilizing transformer training by preventing attention entropy collapse." International Conference on Machine Learning. PMLR, 2023.
> We found this method to be interesting and relevant to our work. Therefore, we additionally compared its performance to our method (see Tables 3 and 4).
> [5] Gao, Peng, et al. "Fast convergence of detr with spatially modulated co-attention." Proceedings of the IEEE/CVF international conference on computer vision. 2021.
> Although this paper addresses convergence, it focuses on detection attention heads and their methodology is not applicable for general Self-Attention transformers. SMCA is designed to work with object queries, a specificity of the Detection Transformer. The method is more comparable to UP-DeTr.
> Accurately describe the amplification effect and how your method mitigates it, providing empirical evidence like the reduced reliance of post-LN layers on residuals.
>
> We did correctly describe the amplification effect. We explained it at the end of the Slow convergence of the Transformer paragraph in our related works section “The Amplification effect is a phenomenon where the training is destabilized by the significant fluctuations of the Transformer encoder outputs between each gradient update. This is caused by the encoder's reliance on its residual layer.”
> In the introduction, we opt to not dive in too much detail to explain the amplification effect as we do it in the related works section.
>
> Additionally, we would like to specify that our method is not a direct technical solution to the amplification effect but rather a better attention search algorithm aimed at finding the attention quicker to reduce the number of steps. As we realized that it was not clearly mentioned in the first version of our manuscript, we improved the description of our method in the last paragraph of the related work section.
> Revise the related work section to verify claims and better contextualize your research against papers [1-5]
> We have revised the related work section to further verify claims and to better contextualize our research against papers [1-5]. As Rev. uUFR noted, this section already provides detailed explanations of related methods. We decided to add [4] to the section, as it has been included in our baselines, and [5] due to its focus on convergence. However, we opted to leave out [1-3], as previously explained, based on a trade-off between conciseness and relevance.

---

> > ### Comment · Reviewer_KvKK · 2024-11-05
> > **Computational cost usually means time taken, number of steps is an arbitrary metric. Not sure whether the contribution makes conceptual sense.**
> >
> > Thank you for making explicit the motivation for GauAttention, that is finding a single key per query instead of scoring every feature pair. Why do you think a single key is enough? There are situations where attention might want to pair information from multiple keys, I can't come up with an example for image classification but this has been shown at least in language [1].
> >
> > I am also skeptical about the motivation of reducing steps for convergence instead of time taken. Intuitively, one could reduce steps by combining multiple steps into one step, while not reducing performance gain per unit of time, which is what model developers actually care about. I would also compare to [1, 2] given this. I would include them in the related work section, mentioning that they focus on time taken instead of number of steps, given their high similarity. Omitting them from the related work section in the interest of 'conciseness' does not seem right.
> > Minor: reference to training time table is broken on page 9 last para.
> >
> > Thank you for clarifying that this work does not provide a solution to the amplification effect. I found its discussion in the introduction given this fact unnecessary.
> >
> > Overall, my main concerns about evaluation and contextualisation with respect to related works remain.
> >
> > [1] Geva, Mor, et al. "Dissecting recall of factual associations in auto-regressive language models." arXiv preprint arXiv:2304.14767 (2023).

---

### Review · Reviewer_uUFR · 2024-10-15

**Summary Of Contributions:**

This paper proposes an Attention alternative, called Gaussian-Attention (GauAttention), for speeding up convergence of Transformer training. This work is strongly motivated by the prior work: "Understanding the difficulty of training transformers" Liu et. al., EMNLP 2020. The premise is Transformers are difficult to train because of an "Amplification Effect" in Transformer residual connections, which is separate from vanishing gradients. Rather than introduce an initialization method to fix this effect (as was done in the prior work), this work aims to replace the Attention mechanism with a "structured" variant, where the tokens are more directly learned via a Gaussian learnable sampling module. The intent is to increase convergence rate rather than wall clock time.

The proposed method has parameters per training sample, so extensions of this method instead use embeddings to compress the model. In a separate phase, the model is distilled using multiple losses into a standard Transformer. Evaluations on CIFAR10 and ImageNet compare the convergence of GauAttention models vs. ViT variants.

**Audience:**

Yes

**Claims And Evidence:**

No

**Requested Changes:**

The weaknesses presented above are critical to understanding the paper. For recommendations, it would be helpful to better understand how GauAttention is learning. For example, how sensitive is GauAttention to overfitting or underfitting with the extra parameters it has? It would also be helpful to understand the wall clock and performance implications of GauAttention, aside from convergence.

**Strengths And Weaknesses:**

Strengths:
* The proposed method is interesting and potentially promising for future work.
* Writing is good (though technical details are confusing).
* Section 3.3.3 and Figure 2 were relatively easy to understand.

Weaknesses:
* The paper has spent a lot of time reviewing prior work, but relatively little space is spent explaining the proposed method, which should be at the center of the work. For example, in section 3, Self-Attention is explained for nearly as long as Gaussian-Attention. Specifically, section 3.2 does not state what the purpose of Gaussian-Attention is; the section starts in the technical details without proper motivation "The GauAttention layer process begins by assigning learnable parameters...". The result is the reader is left to piece together what sampling has to do with the attention mechanism. Under normal circumstances, the reader could perhaps piece together the story, but the current presentation briefly states utilized techniques and decisions, making it partially unclear how the method is implemented. There should be space to explain these details as Page 9 was not fully used.
* In "Understanding the difficulty of training transformers" Liu et. al., EMNLP 2020, Section 4 provides a precise definition of the "Amplification Effect" in terms of pre and post LayerNorm Transformers. How does the GauAttention module fit in that analysis? Can we prove GauAttention has different convergence properties at the layer level?
* Figure 1 is not easy to understand (especially as a standalone figure i.e., without reading the entire text). What is a "tokenizer perceptron"? Why do pictures of a dog map to 3 vectors, each with 3 channels? Why do the vectors get combined before being interpolated? The bottom half of the diagram could use more detail.
* The presentation of model variants used in the evaluation felt unnecessarily complicated. For example, compare Table 1 to Table 2 and Table 3: it is difficult to say which is which. Why is ViT_R0 introduced in the first sentence (on the prior page)?
* Table 2 uses training accuracy. It's well understood that training accuracy can be high if a model is overfitting to the data. What makes this case different?
* Results don't report confidence intervals or sensitivity/ablations to hyperparameters. What is the effect of learning rate? How to tune the early stopping? The exception being Table 4 is an ablation on loss weights reported in terms of time to 84% accuracy.
* The method is worse than the baseline ViT_R0 on ImageNet, yet there isn't a convincing explanation for why.
* Figure 4 is not very clear. If the intent is to compare ViT to GauViT Keys, perhaps plotting a correlation directly would be more clear?

---

> ### Author Response · Authors · 2024-11-01
> **Response to Reviewer uUFR**
>
> We thank the reviewer uUFR for their thoughtful feedback, which was carefully considered to improve our work. We also appreciate the positive remarks on our method as "interesting and potentially promising for future work", as well as the comments on the clarity of our figures and writing quality.
>
> Need for clearer motivation for Gau Attention
>
> Thank you for highlighting the need for a clearer explanation of our method's motivation and purpose. We acknowledge that the initial submission lacked a comprehensive introduction to the reasoning behind GauAttention. We revised the introduction, the end of the related works section, and Section 3.2 to provide a more thorough explanation of the intuition and motivation for using Gaussian sampling.
>
> In summary, the GauTransformer is inspired by the observation that traditional self-attention assigns scores to all keys for each query. Rather than comparing each feature pair, we hypothesize that sampling a single key per query and adjusting the sampling distribution to focus on relevant keys can accelerate the search for optimal attention scores. This method uses parameters tailored to each training example, quickly determining the attention for individual instances.
>
> To improve clarity, we have expanded Section 3.2 to more explicitly connect the Gaussian sampling approach with its purpose in attention optimization, following the technical details with clear motivation. We also ensured that the paper utilized the available space effectively to better explain the methodology and decisions involved.
>
> Amplification Effect Analysis
> We clarify that our method is not a direct solution to the amplification effect but rather an improved attention search algorithm designed to find optimal attention more quickly, thereby reducing the required number of steps for convergence. Recognizing that this distinction was not clearly conveyed in the initial manuscript, we have revised the last paragraph of the related work section to better describe our approach. To compare the convergence quality, we show the evolution of the activation of the attention scores in a graph in Figure 4.
>
> Improvement of the Figure 1
> Thank you for your feedback on Figure 1. In the newer manuscript, we have made the following updates to Figure 1 :
> We changed the label “Tokenizer Perceptron Module” to “Linear Projection of Flattened Patches” to better express how ViT works.
> Reserved rounded-corner shapes for outputs and squared-corner shapes for the rest.
> Added a legend.
> Added details to (a) and (b) to enhance comprehension.
> Modified the display of patches and output vector shapes to better convey the idea that there are multiple patches and corresponding vectors.
> Enhance the Figure 1 caption with additional details.
>
> Clarification of the model tables
> Thank you for shedding light on this, we forgot to mention that we used Atto for CIFAR-10 and Tiny for ImageNet. To avoid redundancy in Table 3, we did not specify the model size used as it is the same across models for each dataset. To make this clearer, we adjusted the Model Variants paragraph, specify which model size is used for each dataset in the captions of Table 3 and the new Table 4, and put the full suffix in Table 4.
>
> Confidence levels
> Following this suggestion, we trained 3 seeds for each model on CIFAR-10 to report a more robust evaluation in Table 3. However, we were unable to do the same on ImageNet due to our training budget.
>
> Performance on ImageNet
> To improve the distillation on ImageNet of GauViT, we increased the contribution of the feature distillation loss from 70% to 85%. By doing so we were able to achieve a slight increase of performance over ReZero. We reported the details of the loss adjustments and the new results in the latest version of the manuscript.
>
> Report training time
> We thank you for the suggestion, following the review, we added a training time benchmark table in our manuscript (Table 4).

---

### Author Response · Authors · 2024-11-01
**Global Response**

We sincerely thank the reviewers for their detailed and valuable feedback on our manuscript. We are glad to hear that our method is promising and that our summarization of the practical challenges of the Transformers was well-received. In response to the reviewers' insightful suggestions, we have made several revisions and enhancements, including:
Strengthened motivation behind Gaussian Attention
Improved clarity and detail in Figure 1
A more robust experimental process on CIFAR-10
Enhanced results on ImageNet
Inclusion of an additional baseline for comparison
Expanded Related Works section to include two relevant prior studies
New benchmark data on training time

---

### Decision · Action_Editor_uJeg · 2024-12-24

**Recommendation:** Reject

**Comment:**

All three reviewers still hold concerns after reconsidering the revised version. The paper falls short in the empirical evaluation, the comparison between baselines. It is concerned in the weak motivation and conceptual flaws. Therefore, it is not ready for publication.

**Audience:**

Transformers are of relevance in the deep learning community.

**Claims And Evidence:**

Claims made in the submission, e.g. the superiority of the approach in terms of efficiency, are not well supported by the studies presented in the paper.